# Not All Out-of-Distribution Data Are Harmful to Open-Set Active Learning

**Yang Yang**[1], **Yuxuan Zhang**[1], **Xin Song**[2], **Yi Xu**[3*]
[1]Nanjing University of Science and Technology
[2]Baidu Talent Intelligence Center, Baidu Inc
[3]Dalian University of Technology
{yyang, xuan_zhang}@njust.edu.cn, songxin06@baidu.com, yxu@dlut.edu.cn

## Abstract

Active learning (AL) methods have been proven to be an effective way to reduce the labeling effort by intelligently selecting valuable instances for annotation. Despite their great success with in-distribution (ID) scenarios, AL methods suffer from performance degradation in many real-world applications because out-of-distribution (OOD) instances are always inevitably contained in unlabeled data, which may lead to inefficient sampling. Therefore, several attempts have been explored open-set AL by strategically selecting pure ID instances while filtering OOD instances. However, concentrating solely on selecting pseudo-ID instances may cause the training constraint of the ID classifier and OOD detector. To address this issue, we propose a simple yet effective sampling scheme, Progressive Active Learning (PAL), which employs a progressive sampling mechanism to leverage the active selection of valuable OOD instances. The proposed PAL measures unlabeled instances by synergistically evaluating instances' informativeness and representativeness, and thus it can balance the pseudo-ID and pseudo-OOD instances in each round to enhance both the capacity of the ID classifier and the OOD detector. Extensive experiments on various open-set AL scenarios demonstrate the effectiveness of the proposed PAL, compared with the state-of-the-art methods. The code is available at `https://github.com/njustkmg/PAL`.

## 1  Introduction

Deep learning has achieved great success in many tasks such as image classification [14], article prediction [35], and graph classification [32]. The main factor underlying its excellent performance lies in the availability of an extensive dataset with manual annotations, which can be costly in real-world applications [16]. To reduce the high labeling cost, active learning is an effective approach that involves iteratively selecting valuable instances from the unlabeled data pool and querying their labels for model retraining [4].

The basic assumption shared by various existing active learning methods is that all unlabeled instances are collected from the ID domain [1, 28], showing that the labeled and unlabeled instances share the same distribution. However, this assumption is unrealistic in real-world scenarios as unlabeled instances are primarily gathered from the open world, where massive instances belong to the OOD domain. Taking the image classification task as an example, the goal is to classify mammals, but a considerable proportion of the unlabeled instances consists of unknown classes (i.e., OOD domain), such as bees, apples, and tables. Once these images are selected for labeling, it would be a waste of the annotation budget since they are not used for training the ID classifier. Therefore, it is important

---

*Corresponding author

37th Conference on Neural Information Processing Systems (NeurIPS 2023).

to detect ID and OOD instances before the annotation in open-set AL, which is not applicable in traditional active learning methods. Consequently, the main challenge in open-set AL revolves around effectively selecting valuable ID instances for training classifier and distinguishing the OOD instances.

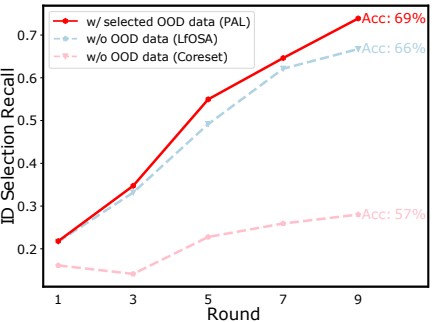

(a) The power of OOD instances.

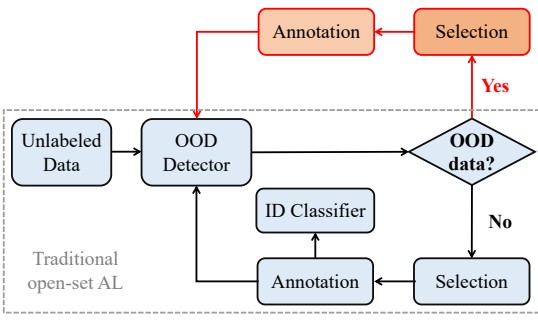

(b) The proposed PAL process

Figure 1: (a) the effectiveness of actively querying valuable pseudo-OOD instances. ID Selection Recall (higher is better): the ratio of selected ID instances to the total number of ID instances. Acc: testing accuracy; w/: with; w/o: without. Experiments are conducted on CIFAR-100 using the first 20 classes as seen classes. We employ (1) the traditional active learning method Coreset; (2) the existing open-set AL approach LfOSA that only selects pseudo-ID instances; and (3) the proposed PAL that actively queries pseudo-OOD instances to enhance the training of the detector and classifier. (b) the distinction between the proposed PAL process and the traditional open-set AL process (in the dashed rectangle). The proposed PAL includes an additional module (the process in red) that carefully leverages detected OOD instances during the model training progress.

To address the challenges of open-set AL, a few studies have proposed [12, 22, 23], which commonly attempt to effectively filter the OOD instances in the unlabeled pool by adopting an extra OOD detector, thus increasing the ID purity of instances in the query set. Their goal is to optimize the selection of ID instances while eliminating OOD instances during each query round, aiming to exclusively add ID instances for model training. In other words, these methods mainly focus on learning the ID classifier, consequently restraining the resolution of the OOD detector. In contrast, we believe that not all OOD instances are harmful, and valuable OOD instances are also significant for both the classifier and detector training. For example, as shown in Figure 1 (a), we observe that actively selecting OOD instances based on their importance (i.e., PAL method) can achieve higher classification accuracy and selection recall, which is attributed to the promotion of ID purity and OOD filtering. Therefore, valuable OOD instances can promptly strengthen the boundary between ID and OOD instances, and further improve the ID purity for classifier retraining.

To this end, we propose a simple yet effective Progressive Active Learning (PAL) scheme that strategically introduces valuable pseudo-ID and pseudo-OOD instances for enhancing both the ID classifier and OOD detector. Different from existing open-set AL methods that solely focus on selecting pseudo-ID instances, our approach goes a step further by actively incorporating valuable pseudo-OOD instances, which allows us to explicitly retrain the OOD detector in conjunction with ID instances, resulting in a mutual promotion of the detector and classifier. The proposed PAL process is illustrated in Figure 1(b). Besides, to comprehensively measure the unlabeled instances, we propose to estimate the informativeness of instances by calculating entropy using the predictions from one-vs-all (OVA) classifiers, while evaluating the representativeness of instances with the automatically learned meta-weights.

## 2 Related Work

**Active Learning.** Active Learning aims to train a classifier by querying labels of the most valuable instances in the unlabeled data, which can greatly reduce the manual labeling cost [7]. The core idea of existing active learning approaches is to ensure that selected instances can significantly improve model performance, and various effective sampling strategies are designed [9, 24, 31, 37]. The initial sampling criteria is uncertainty-based sampling [4], which prefers to select instances that maximally reduce the model uncertainty. For example, [9] proposed an uncertainty measure that generalizes margin-based uncertainty; [37] employed the pairwise gradient length as a metric of informativeness.

Furthermore, various representativeness-based methods are proposed, which prefer to select instances that match the data distribution. For example, [24] estimated the expected reduction in error that directly optimizes expected future error; [1] designed batch active learning with diverse gradient embeddings, which sampled groups of instances that are disparate and high magnitude. However, this family of approaches usually assumes in-distribution settings that the unlabeled instances are drawn from the same distribution of labeled examples, which is not suitable for effectively filtering OOD instances during the query round.

Several recent approaches have attempted to carry out the open-set AL [3, 12, 22, 23]. [3] introduced a contrastive AL framework to learn two contrastive coding models for calculating informativeness and representativeness of an example; [22] introduced an auxiliary network to model the per-example max activation value distribution with a gaussian mixture model, which can dynamically select the instances with the highest probability from known classes in the unlabeled set; [23] employed the meta-learning technique to find the best trade-off between purity and informativeness; [12] selected a set that maximize the distance on entire unlabeled data while minimizing the distance to the identified OOD instances. However, we find that these attempts are limited in filtering OOD instances, as they usually assign greater weight to maintain ID purity, thus causing the learning constraint of the ID classifier and OOD detector. Consequently, it is vital to develop effective approaches that achieve a comprehensive balance between the ID classifier and the OOD detector.

**Out-of-distribution Detection.** OOD detection aims to identify instances outside the target domain [27, 30, 39], i.e., ID domain, which is vital for securing the model's safety and interpretability [15, 34]. Existing methods always calculate instances' confidence scores based on the predictions of the classifier trained on ID instances such as the entropy of the prediction distribution [2, 18, 33]. For example, [18] detected OOD instances with softmax score by considering temperature scaling and input pre-processing; [2] introduced a new model layer OpenMax, which used activation vectors to estimate the probability of deep network failure. To build a more robust detector, several approaches attempted to utilize the generative adversarial network to generate pseudo-OOD instances for assisting learning. For example, [11] proposed task-aware variation adversarial AL that considered data distribution of both label and unlabeled pools; [8] confirmed that generative model accuracy may not always be positively correlated with OOD detection performance. Nonetheless, the advantage of open-set AL lies in the existence of OOD instances within the unlabeled pool, making the fine-grained usage of OOD instances possible.

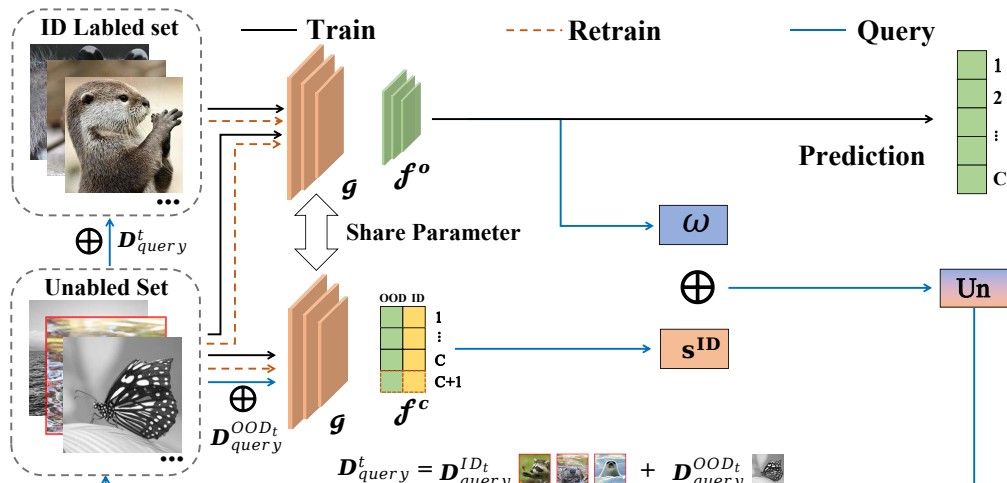

Figure 2: **Framework of the proposed Progressive Active Learning (PAL).** We initialized the ID classifier and OOD detector with the ID labeled data. During the query rounds, we explicitly query both pseudo-ID and pseudo-OOD instances with the designed sampling criterion to enhance the ID classifier and OOD detector.

# 3 Proposed Method

**Problem Setting.** We denote by the labeled data $\mathcal{D}_l = \{(\mathbf{x}_i^l, \mathbf{y}_i^l)\}_{i=1}^{N_l}$, where each $\mathbf{x}_i^l$ belongs to the $C$ known labels $\mathbf{y}_i^l \in \mathbf{R}^C$ and the unlabeled data $\mathcal{D}_u = \{\mathbf{x}_j^u\}_{j=1}^{N_u} = \{\mathcal{D}_u^{ID}, \mathcal{D}_u^{OOD}\}$, where $\mathcal{D}_u^{ID}$ denotes the ID unlabeled data and $\mathcal{D}_u^{OOD}$ represents the OOD unlabeled data, $\mathcal{D}_u^{ID} \cap \mathcal{D}_u^{OOD} = \emptyset$. In other words, an unlabeled instance $\mathbf{x}_j^u$ may belong to an unknown class. In active learning, a human oracle is requested to label query set $\mathcal{D}_{query}^t$ at $t$-th query round, which consists of OOD query set $\mathcal{D}_{query}^{OOD_t}$ and ID query set $\mathcal{D}_{query}^{ID_t}$ (i.e., $\mathcal{D}_{query}^t = \mathcal{D}_{query}^{OOD_t} \cup \mathcal{D}_{query}^{ID_t}$). Considering the restrictions on the annotation budget, the task of traditional open-set AL is to construct the query set that contains more ID instances for assisting ID classifier building.

**Approach Overview.** The framework of the proposed PAL is illustrated in Figure 2, including two key components: 1) importance score considering both informativeness and representativeness; 2) progressive sampling mechanism that precisely manages the ID and OOD instances within the query set. Diverging from traditional open-set AL attempts, we adopt a strategy of actively selecting the valuable pseudo-ID and pseudo-OOD instances to enhance the capacities of the classifier and detector simultaneously. In the following subsections, we first introduce the calculations of the importance score and then give the details of progressive sampling. Finally, we present a theoretical understanding of using OOD instances.

## 3.1 Sampling Criteria

**Uncertainty weight.** To investigate the informativeness, we employ the one-vs-all (OVA) classifiers [26] to learn a boundary between ID and OOD instances. In detail, OVA classifiers include $C$ sub-classifiers, each one is to distinguish whether the instance belongs to the corresponding class or not, e.g., the prediction of $c$-th sub-classifier can be denoted as $\mathbf{p}^c(\mathbf{x}^l) = f^c(g(\mathbf{x}^l)) \in \mathbf{R}^2$, where $g$ represents the feature encoder, $f^c$ denotes the $c$-th classifier. That is to say, $\mathbf{p}^c(\mathbf{x}^l) := (\mathbf{p}^c(p = 1|\mathbf{x}^l), \mathbf{p}^c(p = 0|\mathbf{x}^l))$, where $\mathbf{p}^c(p = 1|\mathbf{x}^l)$ and $\mathbf{p}^c(p = 0|\mathbf{x}^l)$ denotes the probability of instance being an ID and an OOD respectively for class $c$. The overall loss of the OVA classifiers can be formulated as:

$$\ell_{OVA} = -\sum_{i=1}^{N_l}(\log \mathbf{p}^{y_{\mathbf{x}_i^l}}(p = 1|\mathbf{x}_i^l) + \min_{c \neq y_{\mathbf{x}_i^l}} \log \mathbf{p}^c(p = 0|\mathbf{x}_i^l)), \tag{1}$$

where $y_{\mathbf{x}_i^l}$ denotes the class of instance $\mathbf{x}_i^l$. Following [26], we adopt the hard-negative sampling technique for sub-classifiers. The core idea of OVA classifiers is that each sub-classifier outputs a distance representing how far the input is from the corresponding class [26]. Therefore, the sub-classifiers are effective in identifying OOD instances. To be specific, the probability of instance $\mathbf{x}^u$ to be an ID can be calculated by $s^{ID} = 1 + \mathbf{p}^{\hat{c}}(p = 1|\mathbf{x}^u) \log \mathbf{p}^{\hat{c}}(p = 1|\mathbf{x}^u)$, where $\hat{c} = \arg\max_c \mathbf{p}^c(p = 1|\mathbf{x}^u)$. Actually, $s^{ID}$ can express the informativeness of a given unlabeled instance, and larger $s^{ID}$ represents more likely the instance to be an ID instance.

**Meta-weight.** In addition to $s^{ID}$, we design a representativeness criterion by learning the meta-weight of each unlabeled instance. Since the OOD instance in unlabeled set leads to a distribution mismatch, inspired by [5], we aim to automatically learn the weight of each unlabeled instance via tracking the effect of the supervised learning model to prevent OOD instances, which can be formulated as the following bi-level optimization problem:

$$\min_{\omega} \sum_{i=1}^{N_l} \ell_{sup}(\mathbf{x}_i^l, \mathbf{y}_i^l, \hat{\Theta}(\omega)),$$

$$s.t. \quad \hat{\Theta}(\omega) = \arg\min_{\Theta} \sum_{i=1}^{N_l} \ell_{sup}(\mathbf{x}_i^l, \mathbf{y}_i^l, \Theta) + \sum_{j=1}^{N_u} \omega_j \ell_{un}(\mathbf{x}_j^u, \Theta), \tag{2}$$

where $\Theta = (\Theta_1, \Theta_2)$, $\ell_{sup}(\mathbf{x}_i^l, \mathbf{y}_i^l, \Theta) = -\sum_{c=1}^C \mathbf{y}_i^l \log f_{\Theta_1}^{o,c}(g_{\Theta_2}(\mathbf{x}_i^l))$ is the supervised loss, $\ell_{un}(\mathbf{x}_j^u, \Theta) = -\sum_{c=1}^C f_{\Theta_1}^{o,c}(g_{\Theta_2}(\mathbf{x}_j^u)) \log f_{\Theta_1}^{o,c}(g_{\Theta_2}(\mathbf{x}_j^u))$ is the unsupervised loss, and $f^o$ denotes the ID classifier. The problem (2) captures the process of learning the optimal model parameter $\hat{\Theta}(\omega)$

through empirical risk minimization using labeled and weighted unlabeled instances, the model is then evaluated on $\mathcal{D}_l$ to optimize the weights, aiming to enhance the reliable performance. Thus, the learned $\omega$ is considered as the measurement of representativeness, with higher values indicating that ID instances are more representative.

## 3.2 Progressive Sampling

Open-set AL aims to precisely select as many ID instances as possible from the unlabeled pool. With the $s^{ID}$ and $\omega$, we select the first $b$ instances with the highest probability as the query set:

$$un(\mathbf{x}^u) = \omega + \mu s^{ID} \tag{3}$$

where $\mu$ is the hyper-parameter. However, unlike existing open-set AL attempts that only concern ID selection, we empirically find that valuable OOD can enhance the detector and further boost ID selection. Therefore, we propose a simple yet effective progressive sampling. To be specific, in each round, we actively select $b$ instances including $N_{query}^{ID}$ pseudo-ID instances and $N_{query}^{OOD}$ pseudo-OOD instances, noting that we select OOD instances with the lowest $un(\mathbf{x}_j^u)$. At the first query, we set $N_{query}^{OOD} \gg N_{query}^{ID}$. After that, the OVA classifiers are extended to $C + 1$ classifiers, and the $un(\mathbf{x}^u)$ is given by

$$un(\mathbf{x}^u) = \begin{cases} \omega + \mu(1 - s^{ID}), & \text{if } \hat{c} = C + 1 \\ \omega + \mu s^{ID}, & \text{otherwise} \end{cases} \tag{4}$$

## 3.3 Classifier Training

Based on the newly labeled data $\mathcal{D}_{query}^t$, we retrain both the ID classifier and OOD detector. In detail, the current ID labeled data can be represented as $\mathcal{D}_l^t = \mathcal{D}_l^{t-1} \cup \mathcal{D}_{query}^{ID_t}$, where $\mathcal{D}_l^{t-1}$ denotes the accumulated ID labeled data, $\mathcal{D}_{query}^{ID_t}$ denotes the ID labeled data from the detected ID data. The ID classifier $f^o$ is learned by minimizing the supervised loss $\sum_{i=1}^{|\mathcal{D}_l^t|} \ell_{sup}(\mathbf{x}_i^l, \mathbf{y}_i^l, \Theta)$. On the other hand, the current ID and OOD data for training OVA classifiers can be denoted as $\mathcal{D}_{l_{all}}^t = \mathcal{D}_{l_{all}}^{t-1} \cup \mathcal{D}_{query}^t$, where $\mathcal{D}_{l_{all}}^{t-1}$ denotes the accumulated ID and OOD labeled data. Notably, in the first round, the number of classes for detector $f^c$ is $C$, and we train $f^c$ with the initial $D_l$. In the subsequent rounds, the number of classes for detector $f^c$ is expanded to $C + 1$ by actively adding the OOD data. In each round, we retrain the $f^c$ according to loss $\ell_{ova}$. The final target of open-set AL is the ID classification, and thereby we can adopt the $f^o$.

## 3.4 Theoretical Understanding

To theoretically understand the use of detected OOD instances in the OOD detector, we first give the following notations. Suppose that the samples $(\mathbf{x}, y) \in (\mathcal{X}, \mathcal{Y})$ follows a unknown distribution $\mathcal{P}$, where $\mathcal{X} = \mathbf{R}^d$ is input space and $\mathcal{Y} = \{0, 1\}$ is output space. The label of ID instance is $y = 0$ and the label of OOD instance is $y = 1$. Let $\ell : \mathbf{R} \times \mathcal{Y} \longrightarrow \mathbf{R}^+$ be the loss of interest. With the input $\mathbf{x}$ and its corresponding label $y$, the expected loss is $\mathcal{L}(f) = \mathbb{E}_{\mathcal{P}}[\ell(f(\mathbf{x}), y)]$. Suppose we have a training dataset $\{(\mathbf{x_1}, y_1), \ldots, (\mathbf{x_n}, y_n)\}$ drawn from distribution $\mathcal{P}$, then the empirical loss is $\widehat{\mathcal{L}}(f) = \frac{1}{n} \sum_{i=1}^n \ell(f(\mathbf{x_i}), y_i)$. Let $C(\mathcal{F})$ be some proper complexity measure of the family of hypothesis class $\mathcal{F}$. We assume that the loss is Lipschitz with constant $L$ and the Rademacher complexity $\mathfrak{R}_n(\mathcal{F}) \leq \sqrt{\frac{C(\mathcal{F})}{n}}$ [10]. To simplify the analysis, we assume the distribution of ID and OOD samples is balanced and the number of OOD samples is the same as the number of ID samples in the training set. We suppose that the number of detected ID instances is $m$ while the number of detected OOD instances is $n - m$ where $m < n$, and among the detected ID instances, the number of real ID instances is $m_0 := \alpha m$ while the number of real OOD instance is $m_1 := (1 - \alpha)m$ with $0 < \alpha < 1$. Then we present the following theorem, whose proof can be found in the supplementary material.

**Theorem 1** *For a Lipschitz loss $\ell$ bounded by $c$, we have the following results with probability at least $1 - \delta$ simultaneously. For the proposed PAL method, the generalization error bound is*

$$\mathcal{L}(f_{PAL}) - \widehat{\mathcal{L}}(f_{PAL}) \le 2L\sqrt{\frac{C(\mathcal{F})}{n}} + c\sqrt{\frac{\log(1/\delta)}{2n}} \le O\left(\frac{1}{\sqrt{n}}\right). \quad (5)$$

*For the standard AL method, the generalization error bound is*

$$\mathcal{L}(f_{AL}) - \widehat{\mathcal{L}}(f_{AL}) \le L\sqrt{\frac{C(\mathcal{F})}{\alpha m}} + L\sqrt{\frac{C(\mathcal{F})}{(1-\alpha)m}} + \frac{c}{2}\sqrt{\frac{\log(2/\delta)}{2\alpha m}} + \frac{c}{2}\sqrt{\frac{\log(2/\delta)}{2(1-\alpha)m}}$$

$$\le O\left(\frac{1}{\sqrt{\alpha m}} + \frac{1}{\sqrt{(1-\alpha)m}}\right). \quad (6)$$

**Remark.** Since $0 < \alpha < 1$ and $n < m$, we know $\frac{1}{\sqrt{n}} \le \frac{1}{\sqrt{\alpha m}} + \frac{1}{\sqrt{(1-\alpha)m}}$, showing that the generalization error bound for PAL method is better than the bound for standard AL method. That is to say, the use of detected OOD instances can improve the effectiveness of the OOD detector.

## 4   Experiments

In this section, we aim to demonstrate the effectiveness of the proposed PAL. Due to the page limitation, more experimental results and details can be found in the supplementary material.

### 4.1   Experimental Setups

**Datasets.** We evaluate the efficiency of PAL on several image classification benchmarks, i.e., CIFAR-10, CIFAR-100 [13] and Tiny-Imagenet [36] datasets following standard open-set AL methods [12, 22]. The CIFAR-10 dataset consists of 50,000 training images and 10,000 test images, with 10 classes and 5,000 images per class in the training set. The CIFAR-100 dataset has 100 classes and 50,000 training images and 5,000 test images, with 500 images per class in the training set. The Tiny-Imagenet dataset consists of 100,000 training images and 10,000 validation images, with 200 classes and 500 images per class in the training set. To construct the open-set ALscenarios, following [22], we set the proportion of ID classes as 20%, 30%, and 40% in experiments. For example, when the proportion is set as 20% on CIFAR-10, CIFAR-100, and Tiny-Imagenet, the first 2, 20, and 40 classes are considered as ID classes and the last 8, 80, and 160 classes are considered as OOD classes.

Table 1: Comparison of testing accuracy (%) on CIFAR-10, CIFAR-100, and Tiny-Imagenet datasets with an ID proportion of 20%. The best results are highlighted in bold, and the second-best results are underlined.

| Datasets | CIFAR-10 | | | | | CIFAR-100 | | | | | Tiny-Imagenet | | | | |
|---|---|---|---|---|---|---|---|---|---|---|---|---|---|---|---|
| Rounds | 1 | 3 | 5 | 7 | 9 | 1 | 3 | 5 | 7 | 9 | 1 | 3 | 5 | 7 | 9 |
| Label Only | 78.6 | | | | | 42.5 | | | | | 21.4 | | | | |
| Random | 88.1 | 93.1 | 95.9 | 96.9 | 97.4 | 44.6 | 49.2 | 52.2 | 54.5 | 56.7 | 22.9 | 26.8 | 32.7 | 37.8 | 39.2 |
| Uncertainty | 88.2 | 93.0 | 96.1 | 97.2 | 97.5 | 44.4 | 49.4 | 49.7 | 54.8 | 55.3 | 21.6 | 28.5 | 35.6 | 37.6 | 41.1 |
| Certainty | 88.1 | 90.8 | 91.6 | 92.4 | 93.0 | 45.0 | 50.7 | 53.1 | 54.4 | 54.9 | 21.7 | 27.9 | 35.4 | 39.8 | 44.7 |
| Coreset | 86.5 | 94.7 | 95.8 | 96.7 | 96.8 | 43.6 | 49.5 | 51.9 | 55.0 | 57.0 | 23.1 | 26.3 | 33.0 | 39.9 | 43.3 |
| BALD | 84.6 | 93.1 | 94.3 | 96.4 | 96.7 | 42.9 | 48.1 | 52.3 | 54.4 | 56.2 | 22.9 | 26.8 | 32.9 | 38.1 | 39.2 |
| OpenMax | 81.3 | 85.8 | 86.6 | 87.0 | 90.7 | 45.0 | 47.3 | 50.1 | 53.9 | 56.0 | 22.0 | 26.1 | 32.2 | 36.9 | 41.9 |
| CCAL | 88.9 | 94.3 | 96.2 | 97.4 | 97.7 | 45.1 | 50.9 | 53.4 | 57.2 | 60.4 | 23.2 | 28.5 | 35.6 | 40.6 | 44.9 |
| MQnet | 88.8 | 94.9 | 96.8 | 97.4 | 97.8 | 45.3 | 51.1 | 57.9 | 59.1 | 61.3 | 23.8 | 28.6 | 35.7 | 42.0 | 45.8 |
| LfOSA | 84.2 | 95.4 | 97.1 | 97.5 | 98.3 | 45.6 | 52.2 | 59.0 | 62.5 | 66.1 | 22.6 | 28.8 | 36.4 | 43.7 | 47.9 |
| PAL | **91.1** | **95.6** | **97.6** | **98.5** | **98.7** | **45.6** | **53.0** | **60.0** | **65.6** | **69.4** | **24.3** | **33.4** | **43.9** | **47.6** | **52.1** |

**Baselines and Performance criteria.** We compare PAL with three different types of methods: 1) supervised method, i.e., Label Only. 2) traditional AL methods, i.e., Random, Uncertainty [17], Certainty [20], Coreset [28], Openmax [2], and BALD [29]. 3) open-set AL methods, i.e., CCAL [3], LfOSA [22] and MQnet [23]. We consider performance from two aspects: 1) ID classification, i.e., classification accuracy. 2) OOD detection during query round, i.e., precision and recall [22].

**Parameter setting.** For all AL methods, following [22], we randomly sample 1%, 10% and 10% of the examples as the initial labeled set on CIFAR-10, CIFAR-100, and Tiny-Imagenet datasets, respectively. To ensure fairness, we employ WideResNet [38] as the backbone for training all methods. Note that the labeled data only contains ID classes. In each AL round, we train the model for 100 epochs, using the SGD optimizer with the momentum parameter of 0.9. The learning rate is initialized as 0.01 with a mini-batch size of 128, and the weight decay is set to be $5 \times 10^{-4}$. Additionally, the annotation budget per query round is limited to 1500 following [22]. All experiments are implemented on a single NVIDIA V100 GPU.

## 4.2 Performance Comparison

Table 1 and Figure 3 record the ID classification and OOD detection results of PAL and other AL methods with the increasing number of queries, where the index $\{1, 3, 5, 7, 9\}$ denotes the query round.

**ID classification.** The results in Table 1 show that: 1) AL methods perform better than the supervised method because the oracle will filter the OOD instances in the query set. However, the promotion is limited due to decreased ID instances, especially in complex datasets such as CIFAR-100 and Tiny-Imagenet. 2) CCAL, LfOSA, and MQnet outperform traditional active learning methods because LfOSA selects a more purified set for querying, including more ID instances and fewer OOD instances. LfOSA performs better than MQnet because MQnet maintains a balance between purity and informativeness of the query set that contains several OOD. 3) PAL consistently exhibits the best performance in each round on all datasets, especially on the Tiny-Imagenet dataset. Compared to other open-set AL methods, PAL improvements about 7.5%, 3.9%, and 4.2% under the 5, 7, and 9 rounds respectively, indicating that PAL could better control the purity ID by constructing a more robust OOD detector.

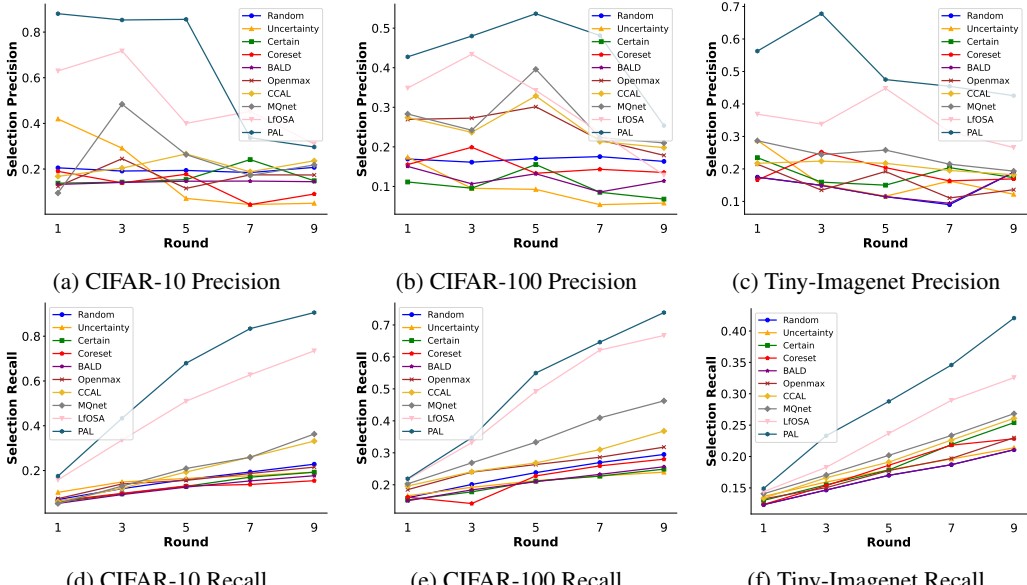

(a) CIFAR-10 Precision  (b) CIFAR-100 Precision  (c) Tiny-Imagenet Precision

(d) CIFAR-10 Recall  (e) CIFAR-100 Recall  (f) Tiny-Imagenet Recall

Figure 3: The precision and recall comparisons of OOD detection on CIFAR-10, CIFAR-100, and Tiny-Imagenet datasets with ID proportion of 20%.

**OOD detection.** The results of Figure 3 reveal that PAL consistently outperforms other baselines in terms of most selection precision and recall. For example, the proposed PAL achieves 15.9% higher precision than LfOSA on Tiny-Imagenet dataset in the last round, which validates that PAL can better eliminate the OOD instances by adopting the discriminative criteria and progressive sampling. Moreover, note that the precision of open-set AL methods shows an initial increase followed by a subsequent decrease, for the reason that we are taking an unreplaced query, so the remaining unlabeled data is gradually reduced. Meanwhile, PAL can select more ID instances at the starting several rounds on CIFAR-10, resulting in fewer ID instances left in the unlabeled set considering the relatively smaller dataset size of CIFAR-10. Thereby, PAL achieves inferior precision compared to LfOSA in later rounds.

To further visualize the validity of pseudo-ID and pseudo-OOD instances, we compare the query purity of PAL with LfOSA in query rounds on three datasets, as illustrated in Figure 4. Although we actively select more OOD instances in the first round, the overall OOD instances are much fewer than LfOSA. Specifically, under the CIFAR-10, LfOSA selected 52.2% OOD instances, whereas PAL only selected 40.3%. Besides, we find that the purities of pseudo-ID and pseudo-OOD are better than LfOSA, i.e., PAL achieves an overall 12.6% higher query purity compared to LfOSA on the Tiny-Imagenet dataset, which validates the effectiveness of progressive building OOD detector.

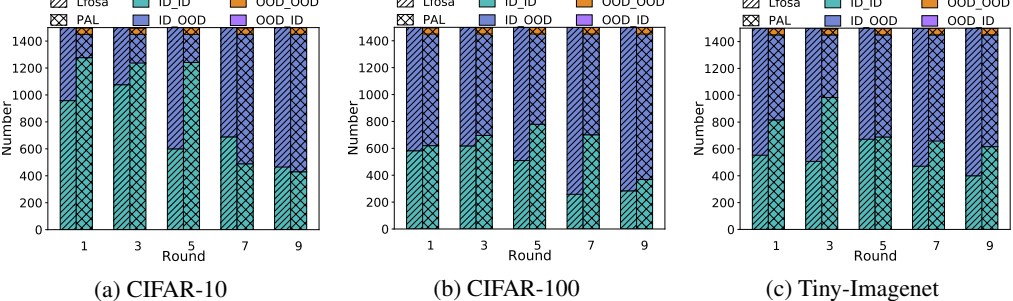

(a) CIFAR-10      (b) CIFAR-100      (c) Tiny-Imagenet

Figure 4: The query purity comparison between LfOSA and PAL. Query purity denotes the ratio of ID_ID instances to the total number of the annotation budget. ID_ID denotes the true positive in the pseudo-ID instances, while ID_OOD represents the false positive in the pseudo-ID instances. Similarly, OOD_ID denotes the false negative in the pseudo-OOD instances, while OOD_OOD represents the true negative in the pseudo-OOD instances. Note that LfOSA only queries the pseudo-ID instances.

## 4.3  Influence of Different ID Class Proportions

In this subsection, we evaluate the influence of different numbers of ID classes. The average results of accuracy, precision, and recall are presented in Figure 5, 6, and 7 respectively. In each figure, the first, second, and third columns represent the results on CIFAR-10, CIFAR-100, and Tiny-Imagenet, respectively, while the first and second rows represent the results with ID proportion as 30% and 40%. Please note that the results of 20% are included in Table 1 and Figure 3. The results exhibit that the proposed PAL always outperforms other methods in terms of ID classification and OOD detection for all cases. For example, when ID proportion is set to 30%, PAL outperforms 3.6%, 3.3%, and 4.3% than the state-of-the-art method, i.e., LfOSA, in terms of accuracy, precision, and recall in the last round on CIFAR-100.

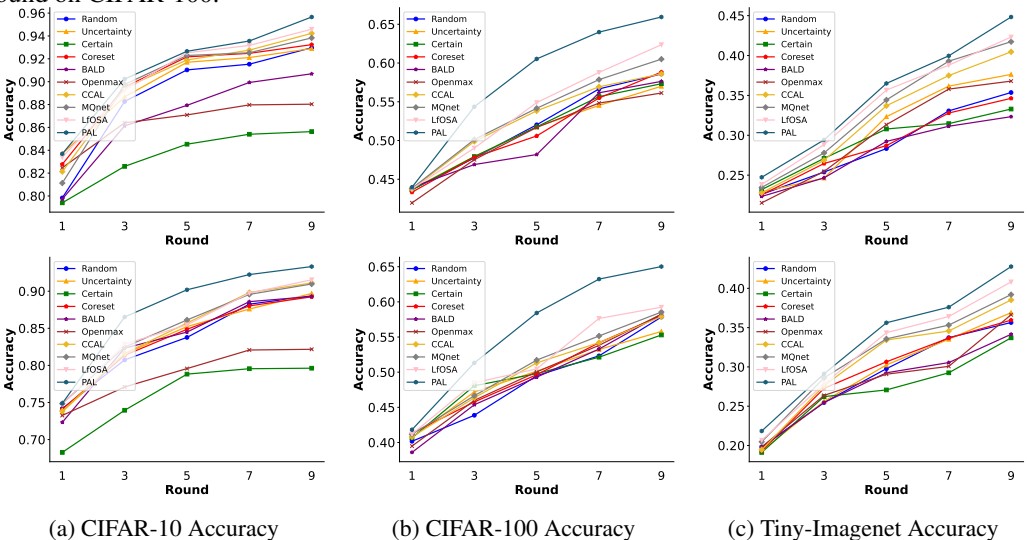

(a) CIFAR-10 Accuracy      (b) CIFAR-100 Accuracy      (c) Tiny-Imagenet Accuracy

Figure 5: Comparison of testing accuracy(%) on CIFAR-10 (first column), CIFAR-100 (second column), and Tiny-Imagenet (third column) datasets, with an ID proportion of 30% (first row) and 40% (second row).

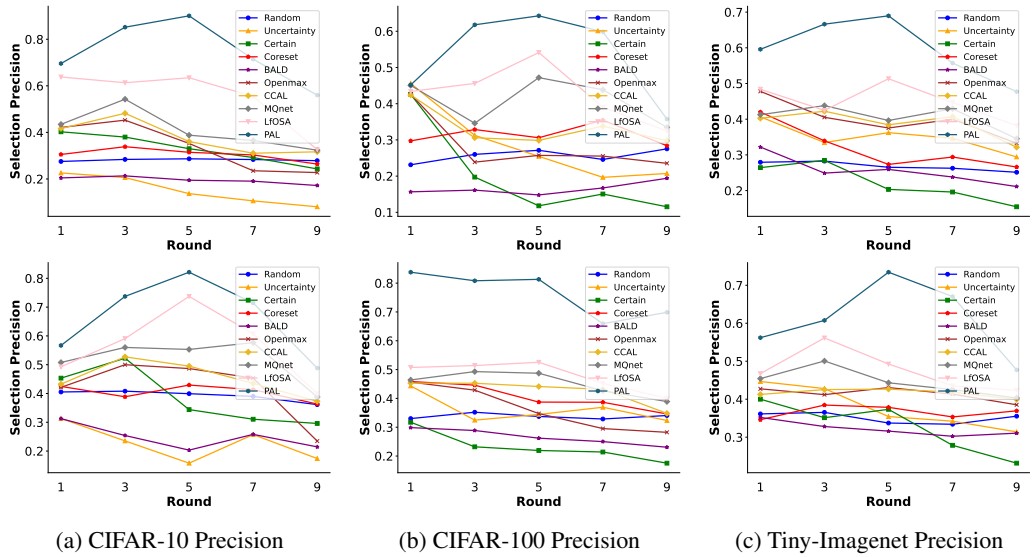

(a) CIFAR-10 Precision      (b) CIFAR-100 Precision      (c) Tiny-Imagenet Precision

Figure 6: The precision comparison of OOD detection on CIFAR-10 (first column), CIFAR-100 (second column), and Tiny-Imagenet (third column) datasets, with an ID proportion of 30% (first row) and 40% (second row).

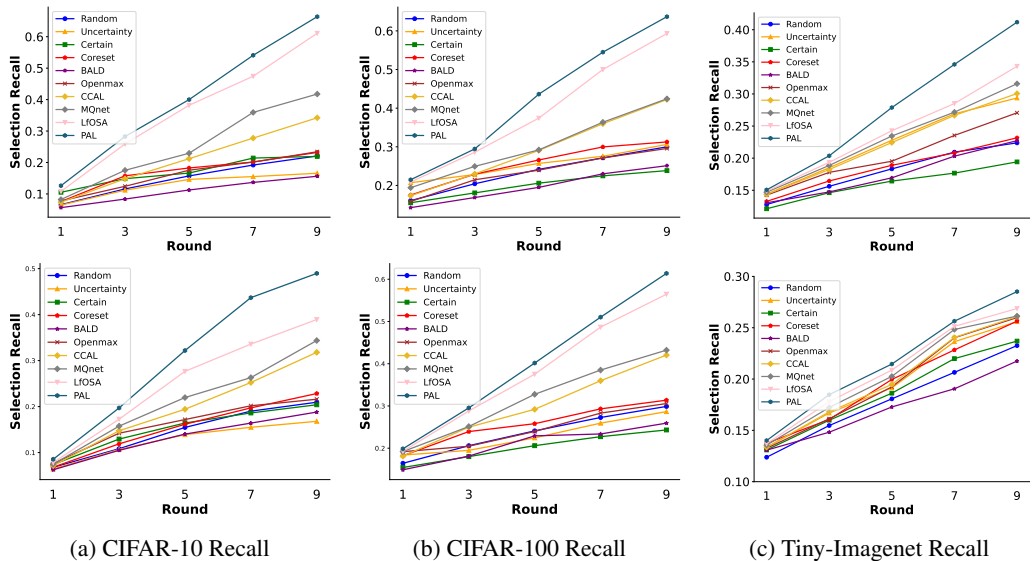

(a) CIFAR-10 Recall      (b) CIFAR-100 Recall      (c) Tiny-Imagenet Recall

Figure 7: The recall comparison of OOD detection on CIFAR-10 (first column), CIFAR-100 (second column) and Tiny-Imagenet (third column) datasets with ID proportion as 30% (first row) and 40% (second row) .

### 4.4 Ablation Study

**Each Component of PAL.** To analyze the contribution of each component of PAL, we conduct ablation study with ID proportion as 20%. The experimental results of classification accuracy are plotted in Figure 8, where w/o $\omega$ denotes the PAL with only the informativeness criterion, and w/o $S^{ID}$ denotes the PAL with only the representativeness criterion. The results indicate that the representativeness criterion has a greater impact on open-set AL in realistic scenarios, for the reason that the representativeness criterion can better distinguish the challenge ID and OOD instances considering the instances' relationships.

**Hyper-Parameter** $\mu$**.** Considering that $s^{ID}$ and $\omega$ can capture different semantics (i.e., informativeness and representativeness) of unlabeled instances, we set $\mu$ with different values, i.e.,

$\{0.2, 0.4, 0.6, 0.8, 1\}$, to empirically investigate the impact of $s^{ID}$ and $\omega$. Table 2 shows the final round accuracy, precision, and recall on CIFAR-100 with ID proportion of 20%. PAL achieves the best ID classification accuracy and OOD detection recall with $\mu = 0.8$, which indicates that PAL can leverage various semantic information of unlabeled instances and effectively improves ID selection by measuring instances' informativeness. Besides, representativeness can better ensure the true positivity of ID instances, allowing PAL to achieve superior precision at $\mu = 0.2$.

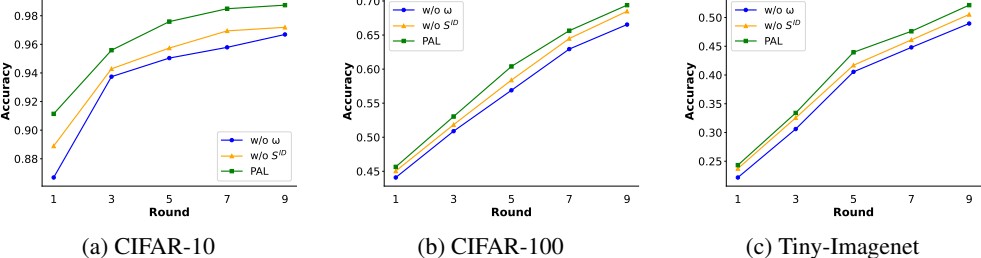

(a) CIFAR-10        (b) CIFAR-100       (c) Tiny-Imagenet

Figure 8: Ablation study conducted on CIFAR-10, CIFAR-100, and Tiny-Imagenet datasets with an ID proportion of 20%.

Table 2: Comparison of accuracy, precision, and recall across different values of $\mu$ in CIFAR-100, with an ID proportion of 20%.

| $\mu$ | 0.2 | 0.4 | 0.6 | 0.8 | 1 |
|---|---|---|---|---|---|
| Accuracy (%) | 66.2 | 67.2 | 68.5 | **69.4** | 67.6 |
| Precision (%) | **32.4** | 27.3 | 25.6 | 25.4 | 17.6 |
| Recall (%) | 67.0 | 69.2 | 71.0 | **73.8** | 69.7 |

**Sampling Criteria.** To evaluate the effectiveness of the proposed sampling criteria, we compare several classic metrics for evaluating representativeness and informativeness. Table 3 records the results on CIFAR-10 and CIFAR-100 datasets, the PAL_Core denotes replacing meta-weight with Coreset [28], which is a traditional metric method used for representativeness. PAL_MSP and PAL_energy methods denote replacing the proposed informativeness criteria with MSP [6] and energy-based [19] criteria respectively. The results exhibit that PAL achieves superior performance on two datasets, revealing the effectiveness of meta-weight in selecting instances.

Table 3: Comparison of testing accuracy (%) for PAL_Core, PAL_MSP, PAL_energy, and PAL on CIFAR-10 and CIFAR-100 with an ID proportion of 20%.

| Datasets | CIFAR-10 | | | | | CIFAR-100 | | | | |
|---|---|---|---|---|---|---|---|---|---|---|
| Rounds | 1 | 3 | 5 | 7 | 9 | 1 | 3 | 5 | 7 | 9 |
| PAL_Core | 87.1 | 93.3 | 96.7 | 97.4 | 98.3 | 44.8 | 49.7 | 59.6 | 63.7 | 65.9 |
| PAL_MSP | 86.0 | 95.1 | 95.2 | 97.5 | 98.0 | 45.1 | 48.4 | 55.8 | 60.1 | 64.7 |
| PAL_energy | 85.6 | 93.7 | 94.7 | 97.0 | 97.9 | 45.32 | 52.5 | 58.6 | 64.5 | 66.2 |
| PAL | **91.1** | **95.6** | **97.6** | **98.5** | **98.7** | **45.6** | **53.0** | **60.0** | **65.6** | **69.4** |

## 5 Conclusion

In this paper, we have proposed PAL, a novel open-set AL method that employs a simple but effective progressive sampling to enhance the performance of OOD detection by actively querying OOD instances at the initial round. PAL included informativeness (i.e., predictions of OVA detector) and representativeness (i.e., automatically learned meta-weights) to measure the importance of unlabeled instances. A theoretical result from the perspective of learning theory revealed that the detected OOD instances enhance the OOD detector. The effectiveness of PAL was demonstrated with extensive empirical results across different scenarios, compared with the state-of-the-art methods. The limitation of the proposed PAL is only applicable to the classification task. In the future, we expect to extend PAL to other computer vision tasks such as object detection.

**Broader Impact.** PAL improves the efficacy of active learning in real-world scenarios. Similar to other active learning methods, it effectively reduces the annotation cost for model training, while the privacy protection issue associated with the practical use of AL on realistic datasets deserves more attention.

## Acknowledgments

The authors thank the anonymous reviewers for their helpful comments. This work is partially supported by National Key RD Program of China (2022YFF0712100), NSFC (62006118, 62276131), Natural Science Foundation of Jiangsu Province of China under Grant (BK20200460), Jiangsu Shuangchuang (Mass Innovation and Entrepreneurship) Talent Program, Young Elite Scientists Sponsorship Program by CAST, the Fundamental Research Funds for the Central Universities (NO.NJ2022028, No.30922010317, DUT No. 82232031).

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

# The Supplementary of Not All Out-of-Distribution Data Are Harmful to Open-Set Active Learning

## A  Proof of Theorem 1

Firstly, we prove the generalization error of the proposed PAL method. Following Theorem 1 of [10], we have

$$\mathcal{L}(f_{PAL}) - \widehat{\mathcal{L}}(f_{PAL}) \leq 2L\mathfrak{R}_n(\mathcal{F}) + c\sqrt{\frac{\log(1/\delta)}{2n}}, \tag{7}$$

where $\mathfrak{R}_n(\mathcal{F})$ is the Rademacher complexity of a function class $\mathcal{F}$ and $n$ is the number of training samples. With the Rademacher complexity $\mathfrak{R}_n(\mathcal{F}) \leq \sqrt{\frac{C(\mathcal{F})}{n}}$, we give equation 5.

Next, we prove the generalization error of the standard AL method. To this end, we denote by $\mathcal{P} = \mathcal{P}(\mathbf{x}|y = 0)$ the conditional probability for ID data and $\mathcal{P}_1 = \mathcal{P}(\mathbf{x}|y = 1)$ the conditional probability for OOD data. Let $\mathcal{L}_j(f)$ denote the loss from class $j \in \{0, 1\}$: $\mathcal{L}_j(f) = \mathbb{E}_{\mathcal{P}_j}[\ell(f(\mathbf{x}), y)]$, and let $\widehat{\mathcal{L}}_j(f)$ denote its corresponding empirical loss. Then by applying the standard analysis for each class $j$ in Theorem 1 of [10], with probability $1 - \delta/2$ we have

$$\mathcal{L}_j(f_{AL}) - \widehat{\mathcal{L}}_j(f_{AL}) \leq 2L\mathfrak{R}_{m_j}(\mathcal{F}) + c\sqrt{\frac{\log(2/\delta)}{2m_j}}, \quad (j = 0, 1). \tag{8}$$

Since the distribution of ID and OOD samples is balanced, we have $\mathcal{L}(f_{AL}) = \frac{1}{2}\mathcal{L}_0(f_{AL}) + \frac{1}{2}\mathcal{L}_1(f_{AL})$ due to the definitions of the loss functions. Similarly, due to the number of OOD samples being the same as the number of ID samples in the training set, we know $\widehat{\mathcal{L}}(f_{AL}) = \frac{1}{2}\widehat{\mathcal{L}}_0(f_{AL}) + \frac{1}{2}\widehat{\mathcal{L}}_1(f_{AL})$. Finally, by applying the union bound, we have

$$\mathcal{L}(f_{AL}) - \widehat{\mathcal{L}}(f_{AL}) \leq L\mathfrak{R}_{m_0}(\mathcal{F}) + L\mathfrak{R}_{m_1}(\mathcal{F}) + \frac{c}{2}\sqrt{\frac{\log(2/\delta)}{2m_0}} + \frac{c}{2}\sqrt{\frac{\log(2/\delta)}{2m_1}}. \tag{9}$$

With the Rademacher complexity $\mathfrak{R}_{m_j}(\mathcal{F}) \leq \sqrt{\frac{C(\mathcal{F})}{m_j}}$ for $j = 0, 1$, we complete the proof of equation 6 by plugging in $m_0 = \alpha m$ and $m_1 = (1 - \alpha)m$.

## B  Additional Implementation Details

**Implementations.** In the first round, to enhance the effectiveness of OOD detection, we set $N_{query}^{ID} = 300$ and $N_{query}^{OOD} = 1200$. To reduce the influence of the pseudo-OOD budget for the ID classifier, we set $N_{query}^{ID} = 1450$ and $N_{query}^{OOD} = 50$ in the later rounds.

Table 4: Dataset Construction.

| Dataset | Number of ID classes | ID : OOD Ratio | Number of labeled data | Number of unlabeled data | Query size b |
|---|---|---|---|---|---|
| CIFAR-10 (20%) | 2 | 2 : 8 | 100 | 49900 | 1500 |
| CIFAR-10 (30%) | 3 | 3 : 7 | 150 | 49850 | 1500 |
| CIFAR-10 (40%) | 4 | 4 : 6 | 200 | 49800 | 1500 |
| CIFAR-100 (20%) | 20 | 20 : 80 | 1000 | 49000 | 1500 |
| CIFAR-100 (30%) | 30 | 30 : 70 | 1500 | 48500 | 1500 |
| CIFAR-100 (40%) | 40 | 40 : 60 | 2000 | 48000 | 1500 |
| Tiny-Imagenet (20%) | 40 | 40 : 160 | 2000 | 48000 | 1500 |
| Tiny-Imagenet (30%) | 60 | 60 : 140 | 3000 | 47000 | 1500 |
| Tiny-Imagenet (40%) | 80 | 80 : 120 | 4000 | 46000 | 1500 |

**Description for Datasets.** In Section 4, we provide a brief introduction to the construction of the open-set AL scenario. In order to provide a more detailed explanation of the dataset construction, we use CIFAR-100 with the ID proportion of 20% as a specific example. We consider the first 20 classes

as the ID classes. Then, we randomly sample 50 instances from each ID class to form the labeled set $\mathcal{D}_l$, while the remaining instances are unlabeled data $\mathcal{D}_u$. As a result, the labeled and unlabeled sets comprise 1,000 and 49,000 instances respectively, as shown in Table 4.

**Description for baselines.** Here's a brief introduction to the comparison methods in the experiment.

- **Random** introduced randomly selecting instances from the unlabeled pool for labeling.
- **Uncertainty** [17] employed entropy to evaluate the uncertainty of instances.
- **Certainty** [20] considered utilizing the weakly labeled data to query instances based on entropy of local marginals.
- **Coreset** [28] proposed core-set selection with greedy k-center clustering to minimize the maximum distance between labeled data and unlabeled data.
- **BALD** [29] employed dropout as an approximation for Bayesian inference to facilitate active sampling.
- **Openmax** [2] introduced the activation vector to measure whether a instance belongs to the known classes.
- **CCAL** [3] introduced a framework of contrastive active learning, with two contrastive coding models employed to assess the informativeness and representativeness of each example.
- **MQnet** [23] utilized meta-learning techniques to discover an optimal balance between purity and informativeness.
- **LfOSA** [22] leveraged an auxiliary network equipped with a gaussian mixture model to capture the distribution of maximum activation values for each instance, enabling adaptive selection of high-probability instances from known classes.

## C   Additional Experimental Results

**Impact of the annotated OOD_ID data.** Actually, the method of only adding additional data into OOD Detector, leading the annotated OOD_ID data (the annotated ID data after selection of the detected OOD data) not used in ID classifier of PAL. In this section, we aim to empirically investigate the impact of annotated OOD_ID data. To this end, we modify the proposed PAL by adding the annotated OOD_ID data into ID classifier and naming it as PAL+, which is illustrated in Figure 9.

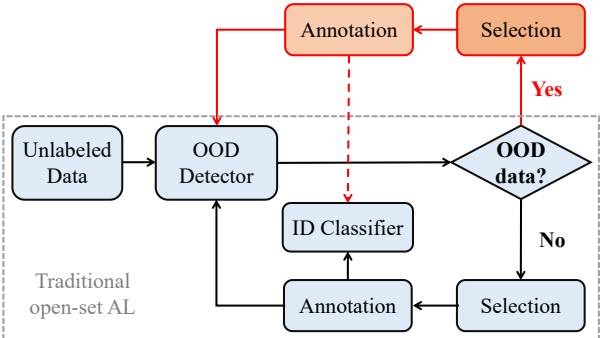

Figure 9: PAL+: PAL with the annotated OOD_ID data added to ID classifier (red dash line).

Considering experimental completeness, we present the classification and OOD detection performance without and with using the OOD_ID instances (i.e. PAL and PAL+), respectively. Note that the experiments are performed on three datasets with an ID proportion of 20%. The results in Figure 10 demonstrate that the proposed PAL and PAL+ are comparable, showing that the annotated OOD_ID instances have minimal impact on the overall performance of the model.

In addition, Table 5 lists the number of annotated OOD_ID instances with their proportion in the detected OOD data, for round 1, 3, 5, 7, and 9 on three data sets. The results in Table 5 show that the annotated OOD_ID instances have a very small proportion in the detected OOD instances, and this may also explain why the PAL and PAL+ are comparable.

Table 5: The number (with its proportion) of annotated OOD_ID instances used in PAL+.

| Seed | Dataset | Round | | | | |
|------|---------|-------|------|------|------|------|
| | | 1 | 3 | 5 | 7 | 9 |
| seed=1 | CIFAR-10 | 5 (10%) | 1 (2%) | 0 (0%) | 0 (0%) | 1 (2%) |
| | CIFAR-100 | 5 (10%) | 5 (10%) | 1 (2%) | 0 (0%) | 0 (0%) |
| | Tiny-Imagenet | 3 (6%) | 3 (6%) | 2 (4%) | 1 (2%) | 0 (0%) |
| seed=2 | CIFAR-10 | 3 (6%) | 0 (0%) | 0 (0%) | 0 (0%) | 0 (0%) |
| | CIFAR-100 | 4 (8%) | 2 (4%) | 1 (2%) | 1 (2%) | 0 (0%) |
| | Tiny-Imagenet | 6 (12%) | 3 (6%) | 0 (0%) | 0 (0%) | 1 (2%) |
| seed=3 | CIFAR-10 | 1 (2%) | 0 (0%) | 0 (0%) | 0 (0%) | 0 (0%) |
| | CIFAR-100 | 1 (2%) | 2 (4%) | 1 (2%) | 1 (2%) | 0 (0%) |
| | Tiny-Imagenet | 2 (4%) | 3 (6%) | 2 (4%) | 0 (0%) | 2 (4%) |

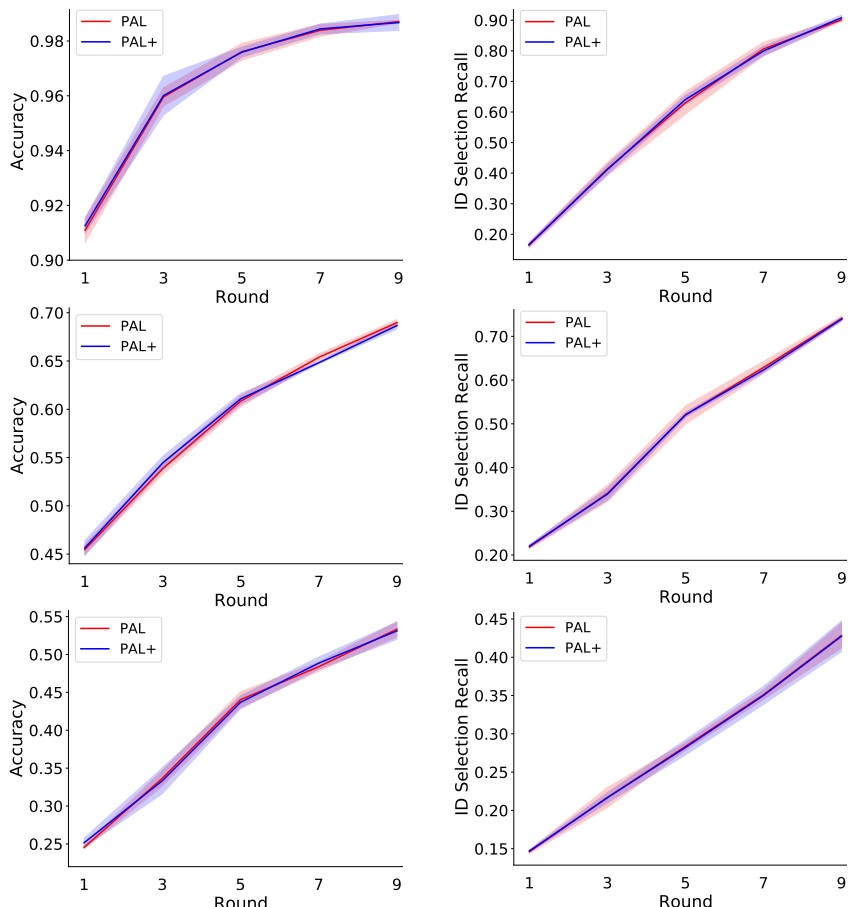

Figure 10: Comparison of classification and OOD detection performance on CIFAR-10 (first row), CIFAR-100 (second row) and Tiny-Imagenet (third row) with an ID proportion of 20%.

**Visualization.** To demonstrate the distribution of selected ID instances, we provide visual results of the iterative query round. In detail, we visualize the feature representations of the labeled and queried correct ID instances via t-SNE [21] on the CIFAR-10 dataset with an ID proportion of 40%. Note that green crosses represent the queried ID instances. Figure 11 records the visualizations of PAL and state-of-the-art methods, i.e., Coreset and LfOSA. The figures reveal that PAL selects more ID instances than other methods, i.e., higher purity in different rounds. Moreover, from the perspective of data distribution, PAL would balance the informativeness and representativeness criteria to select meaningful ID instances for ID classifier training.

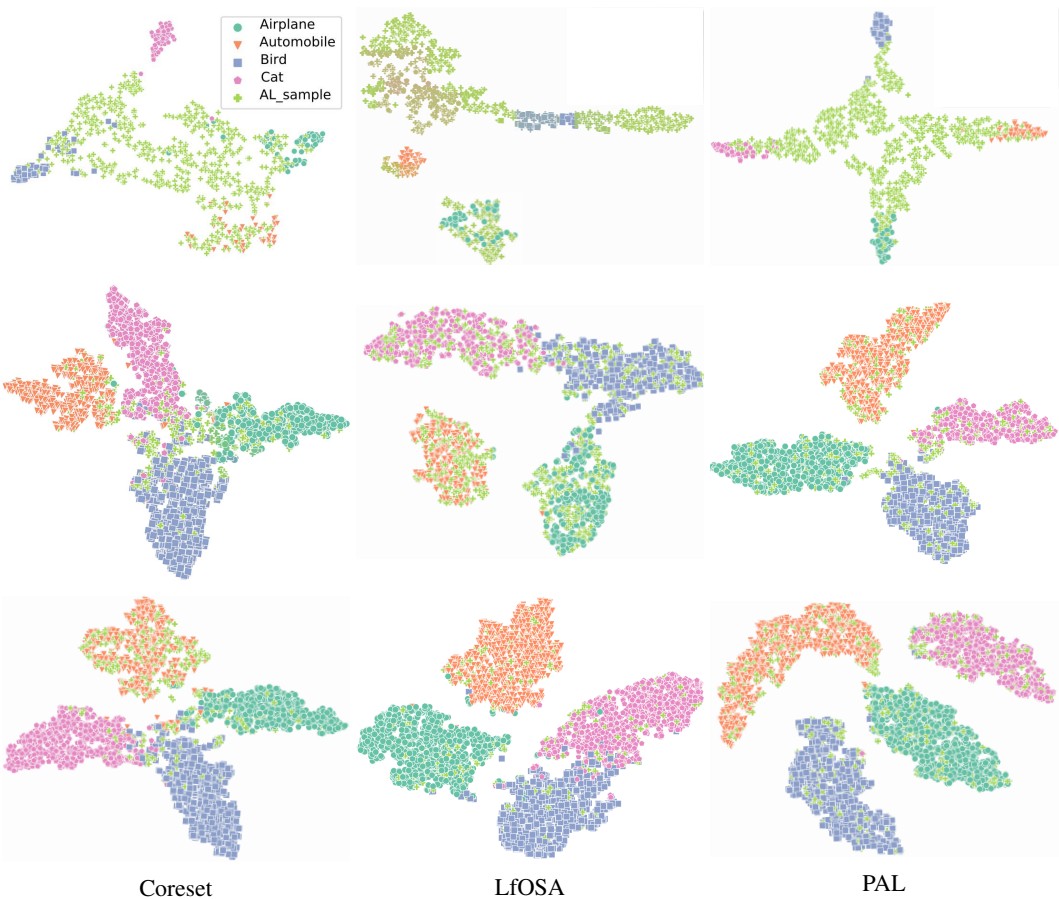

| Coreset | LfOSA | PAL |

Figure 11: The t-SNE visualizations of Coreset, LfOSA, and PAL, considering the 1st (top row), 5th (middle row), and 9th (bottom row) rounds.

**Compare with Open-Set Semi-Supervised Learning.** Although both open-set/safe semi-supervised learning and open-set AL aim to construct a more effective ID classifier by reasonably utilizing unlabeled data, the essential difference lies in whether or not manual participation is necessary. Open-set/Safe semi-supervised learning aims to use all the pseudo-ID instances in unlabeled data according to the model's prediction, whereas open-set AL aims to select the most valuable pseudo-ID instances in unlabeled data for manual labeling to retrain the model without using all the unlabeled data. To evaluate the classification performance of PAL and open-set semi-supervised learning methods, we have compared PAL with two open-set semi-supervised learning methods, DS$^3$L [5] and OpenMatch [25] using the CIFAR-10 and CIFAR-100 datasets with an ID proportion of 20%. As illustrated in Table 6, the results show the proposed PAL also performs better than the open-set semi-supervised learning methods.

Table 6: Comparison of testing accuracy (%) for DS$^3$L, OpenMatch, and PAL on CIFAR-10 and CIFAR-100 with an ID proportion of 20%.

| Method | CIFAR-10 | CIFAR-100 |
|:---:|:---:|:---:|
| DS$^3$L | 74.2 | 48.1 |
| OpenMatch | 82.1 | 66.5 |
| PAL | **98.7** | **69.4** |

**Runtime Cost.** To verify the runtime cost of the proposed PAL, we evaluate it against several baselines, including DS$^3$L, OpenMatch, Coreset, and LfOSA on CIFAR-10 and CIFAR-100 with the ID proportion of 20%. As demonstrated in Table 7, the results show that PAL has the second best runtime cost, outperforming DS$^3$L, OpenMatch, and LfOSA, but falling behind Coreset.

Table 7: Comparison of runtime cost for PAL and other baselines on CIFAR-10 and CIFAR-100 with an ID proportion of 20%.

| Method | CIFAR-10 | CIFAR-100 |
|--------|----------|-----------|
| DS$^3$L | 26.8h | 43.4h |
| OpenMatch | 37.7h | 68.3h |
| Coreset | 2.8h | 3.6h |
| LfOSA | 66.2h | 77.8h |
| PAL | 3.4h | 4.4h |

