# OpenReview forum: "Not All Out-of-Distribution Data Are Harmful to Open-Set Active Learning"
_NeurIPS.cc/2023/Conference — NeurIPS 2023 poster_

### Official Review · Reviewer_ACvU · 2023-06-23

**Soundness:** 2 fair
**Presentation:** 2 fair
**Contribution:** 1 poor
**Rating:** 5
**Confidence:** 4

**Summary:**

This paper proposes an active learning approach for open-set learning. The proposed approach does not try to avoid sampling OOD data as some of the prior work did. Instead, it samples some of the OOD data intentionally to enhance the OOD detector.

**Strengths:**

1.The proposed idea is easy to implement and understand.
2.The proposed problem, open-set active learning, is meaningful to me.


**Weaknesses:**

1.The overall framework is trivial to me. It is a traditional active learning method adapted to an OOD detector and an ID learner.
2.The essential part of active learning, which is the design of the sampling criterion, is neither novel nor plausible to me. For the uncertainty weight, it is merely the prediction score of the model proposed by [27]. What is more, this score tends to approximate the likelihood of a data point being an in-distribution sample. But I do not think this score reveals the informativeness/ sample importance in active learning. Different data points can have very different contributions to learning, the proposed sampling score does not seem to have the ability to differentiate them. The meta-weight can be seen as a loss-based active sampling criterion for an OOD detector, which is trivial to me.
3.All the proposed methods are empirical, there is no quantitative analysis to support the effectiveness of the proposed methods.


**Questions:**

None

---

> ### Author Rebuttal · Authors · 2023-08-09
>
> Q1: "...framework is trivial to me..."
>
> A1: Open-set active learning aims to strategically select pure ID data and filter out OOD data, and thus a powerful OOD detector is important. The common practice of traditional open-set active learning methods, as shown in Figure 1 (b) in the manuscript, always relies on ID data to construct the OOD detector. Subsequently, these methods primarily focus on selecting pseudo-ID instances, which would lead to the weakening of the OOD detector due to the lack of active selection of effective OOD data and affects the ID purity during the selection process.  To address this challenge, we propose a simple yet effective sampling scheme that progressively selects pseudo-ID and pseudo-OOD instances in each round, especially adding valuable OOD data in the initial round. Thereby, it could enhance the capacity of the OOD detector and simultaneously promote the ID classifier by increasing ID purity. Besides, we also provide a theoretical analysis to support why the proposed PAL is better than traditional AL. Specifically, the theoretical results show that the proposed PAL has a better generalization error bound than traditional AL, showing the effectiveness of the proposed PAL. Please refer to **Global Response** for more details about the theoretical analysis.
>
>
> Q2: "... the uncertainty weight, ... the meta-weight..."
>
> A2: Sorry for the confusion. We represent the informativeness of samples by measuring the uncertainty of the most confident ID class. This is accomplished by utilizing the prediction of the OVA classifier. Besides, considering the divergent distributions of ID and OOD data inspired by [Ref 3,  Ref 4], we utilize meta-weight to measure the representativeness of each sample by assessing its similarity to the established data distribution. Instances with higher meta-weights are deemed to be more representative as they match better with the existing data distribution. Besides, following the comments of Reviewer oop5, we replaced the meta-weight with Coreset (we denote the new method by Core_PAL) to validate the effectiveness of the meta-weight by conducting experiments on CIFAR-10 and CIFAR-100 datasets. The results are listed in the following table and show that the Core_PAL performs worse than PAL, revealing the effectiveness of meta-weight in selecting instances.
>
> Table 2: Comparison of classification accuracy (%) for Core_PAL and PAL on CIFAR-10 and CIFAR-100 with an ID proportion of 20%.
>
> |          | CIFAR-10 |       |       |       |       | CIFAR-100 |       |       |       |       |
> | :------: | :------: | :---: | :---: | :---: | :---: | :-------: | :---: | :---: | :---: | :---: |
> |  Round   |    1     |   3   |   5   |   7   |   9   |     1     |   3   |   5   |   7   |   9   |
> | Core_PAL |  87.09   | 93.29 | 96.74 | 97.44 | 98.29 |   44.79   | 49.75 | 59.59 | 63.74 | 65.99 |
> |   PAL    | **91.14** | **95.59** | **97.59** | **98.49** | **98.74** | **45.65** | **53.04** | **60.04** | **65.64** | **69.39** |
>
> [Ref 3] Hugo Larochelle et al., OOD-MAML: Meta-Learning for Few-Shot Out-of-Distribution Detection and Classification, NeurIPS2020
>
> [Ref 4] Cesar Almecija et al., Uncertainty-Aware Meta-Learning for Multimodal Task Distributions, ICLR2023
>
>
> Q3: "...proposed methods are empirical, there is no quantitative analysis..."
>
> A3: In fact, we don't fully understand this question. We will try our best to answer this question, and any additional discussion is welcome. First, we have provided a theoretical analysis (please refer to **Global Response** for details) to show the effectiveness of the proposed PAL. Additionally, we have performed a repeatability experiment in Appendix B. As shown in Figure 10 in the supplementary material, the central line represents the mean value, while the upper and lower limits correspond to its standard deviations for each color. The results demonstrate that our PAL method exhibits consistently high stability.

---

> > ### Comment · Reviewer_ACvU · 2023-08-15
> > **Thank you for the response**
> >
> > Thank you for the response.
> >
> > The rebuttal addresses most of my concerns. I have raised my score accordingly.

---

> > > ### Author Response · Authors · 2023-08-16
> > >
> > > Thank you for your helpful comments and feedback. Please let us know if there are further confusions/questions. We are happy to clarify and try to address them.

---

### Official Review · Reviewer_k1sx · 2023-06-30

**Soundness:** 3 good
**Presentation:** 2 fair
**Contribution:** 3 good
**Rating:** 6
**Confidence:** 5

**Summary:**

This manuscript studies open-set active learning and points out that concentrating solely on selecting pseudo-ID instances may cause the training imbalance of the ID classifier and OOD detector. To address this issue, this manuscript proposes a simple yet effective sampling scheme, dubbed Progressive Active Learning (PAL). Extensive experiments on various open-set AL scenarios demonstrate the effectiveness of PAL.

**Strengths:**

1.This manuscript points out that concentrating solely on selecting pseudo-ID instances may cause the training imbalance of the ID classifier and OOD detector and proposes Progressive Active Learning (PAL) to solve the problem.

2.Extensive experiments on various open-set AL scenarios demonstrate the effectiveness of PAL.

**Weaknesses:**

1.The manuscript lacks corresponding theoretical analysis.

2.The manuscript lacks an introduction to related work on open-set semi-supervised learning and safe semi-supervised learning, such as [1], [2], and [3].

3.In line 39-41, “the main challenges in open-set AL revolve around effectively selecting valuable ID instances for classifier training and distinguishing the OOD instances”, Similar viewpoints have already been proposed by [2].

4.The framework in Fig. 2 is confusing and the flow description is not clear enough. For example, how is f^c trained?

5.In every query, how are N^{OOD}_{query} and N^{ID}_{query} set up?

6.The description of the retraining process is not clear. What is the final loss? Is f^c used for the final testing?

7.In line 136, entropy is used for obtaining s^{ID}. Has the author tried other metrics such as MSP in [4] or energy in [5]?

8.The manuscript lacks important baselines, such as [1], [2], [3]. The manuscript needs to be compared with open-set semi-supervised learning and safe semi-supervised learning methods.

9.The manuscript mentions the need for multiple rounds of query and training, so the runtime of the manuscript needs to be compared with the baseline.

[1] Safe deep semi-supervised learning for unseen-class unlabeled data.
[2] Safe-Student for Safe Deep Semi-Supervised Learning with Unseen-Class Unlabeled Data.
[3] Openmatch: Open-set semi-supervised learning with open-set consistency regularization.
[4] A Baseline for Detecting Misclassified and Out-of-Distribution Examples in Neural Networks.
[5] Energy-based out-of-distribution detection.

**Questions:**

How is N^{ID}_{query} selected? Is the sample with maximum of un(x^u_j) chosen into N^{ID}_{query}? The sample with maximum of un(x^u_j) indicates strong certainty in these samples. Perhaps using model-generated pseudo-labels would be accurate enough, without the need to waste resources on querying these relatively certain samples.

**Limitations:**

Please see Weaknesses.

---

> ### Author Rebuttal · Authors · 2023-08-09
>
> Due to the space limit, we put all tables in the **attached one-page PDF**.
>
> Q1: "...lacks corresponding theoretical analysis..."
>
> A1: We have provided a theoretical analysis in the **Global Rebuttal** to support the proposed PAL method. We theoretically show that PAL has a better generalization error bound than the traditional AL method, which means PAL has better generalization ability than traditional AL. We will include more comprehensive theoretical details in the revision.
>
> Q2: "... lacks an introduction to related work on open-set semi-supervised learning and safe..."
>
> A2: Although both open-set/safe semi-supervised learning and open-set active learning aim to construct a more effective ID classifier by reasonably utilizing unlabeled data, the essential difference lies in whether or not manual participation is necessary. Open-set/Safe semi-supervised learning aims to use all the pseudo-ID instances in unlabeled data according to the model's prediction, whereas open-set active learning aims to select the most effective pseudo-ID instances in unlabeled data for manual labeling to retrain the model without using all the unlabeled data.  We will provide more discussion in the Related Work section.
>
> Q3: "...Similar viewpoints have already been..."
>
> A3: Open-set active learning aims to strategically select pure ID data and filter out OOD data, and thus a powerful OOD detector is important. The common practice of traditional open-set active learning methods, as shown in Figure 1 (b) in the manuscript, always relies on ID data to construct the OOD detector and then concentrates solely on select pseudo-ID instances, which would lead to the weakening of the OOD detector due to the lack of active selection of effective OOD data and affects the ID purity during the selection process. To address this challenge, we propose a simple yet effective sampling scheme that is progressively selecting pseudo-ID and pseudo-OOD instances in each round, especially adding valuable OOD data in the initial round. Thereby, it could enhance the capacity of the OOD detector and simultaneously promote the ID classifier by increasing the ID purity. Besides, we also provide a theoretical analysis to support why the proposed PAL is better than traditional AL. Specifically, the theoretical results show that the proposed PAL has a better generalization error bound than traditional AL, showing the effectiveness of the proposed PAL. Please refer to **Global Response** for more details about the theoretical analysis.
>
> Q4: "... the flow description is not clear enough...how is $f^c$ trained? "
>
> A4: Sorry for the confusion. Specifically, in the first round, the number of classes for detector $f^c$ is C, and we train $f^c$ with the initial $D_l$. In the subsequent rounds, the number of classes for detector $f^c$ is expanded to C+1 by actively adding the OOD data. In each round, we retrain the $f^c$ according to $l_{ova}$ loss with $D_{l_{\text{all}}}^t$. The algorithmic pseudocode is included in Appendix A.
>
> Q5: "...$N_{query}^{OOD}$ and $N_{query}^{ID}$ set up..."
>
> A5: Thanks. In our experiments, the label budget is set to be 1500 in each round following [Ref 1, Ref 2], i.e., $ |D_{query}^{t}| = N_{query}^{OOD_{t}} + N_{query}^{ID_{t}}=1500$. Specifically, in the first round, to enhance the effectiveness of OOD detection, we set $N_{query}^{ID} = 300$ and $N_{query}^{OOD} = 1200$. In the following rounds, to select more valuable ID instances, we set $N_{query}^{ID} = 1450$ and $N_{query}^{ID} = 50$. From Figure 4 in the manuscript, we can conclude that the query purity of PAL is superior to the comparison method, which reveals that we can autonomically select useful pseudo-OOD and pseudo-ID according to our assignment.
>
>
>
> [Ref 1] Pan Du et al., Contrastive active learning under class distribution mismatch, ICCV2021
>
> [Ref 2] Kun-Peng Ning et al., Active Learning for Open-set Annotation, CVPR2022
>
> Q6: "...retraining process is not clear..."
>
> A6: Sorry for the confusion. Actually, the final loss is described in line 174. The final target of open-set active learning is the ID classification, and thereby we can adopt the $f^o$. Besides, we also give the detection result for supplementary validation using $f^c$. We will modify the descriptions in the revision.
>
> Q7: "...tried other metrics such as MSP in [4] or energy in [5]..." and  "...lacks important baselines, such as [1], [2], [3]..."
>
> A7: We have replaced $s^{ID}$ with the MSP/energy-based method and compared PAL with D3SL and OpenMatch. We conducted experiments on CIFAR-10 and CIFAR-100 with the ID proportion of 20%. The results are presented in **Table 3 and Table 4 of the attached one-page PDF** . Furthermore, by employing the official code of D3SL and conducting hyperparameter tuning with values such as meta_lr={6e-5, 3e-4, 2e-3}, un_batch_size={100, 128, 256}, and lr_weight={1e-3, 1e-2, 3e-4}, the results showed in Table 4 signify the most favorable outcomes attained. The results in Table 3 show that by using $s^{ID}$ metric, PAL achieves the best performances on CIFAR-10 and CIFAR-100 datasets, compared with PAL with MSP/energy. The results in Table 4 show that PAL outperforms D3SL and OpenMatch on CIFAR-10 and CIFAR-100 datasets.
>
> Q8: "...the runtime of the manuscript needs to be compared with the baseline."
>
> A8: We have included a comparison of running times for PAL and other baselines on CIFAR-10 and CIFAR-100 with an ID proportion of 20% in **Table 5 of attached one-page PDF** . The results show that PAL has the second fast runtime, which is faster than D3SL, OpenMatch and LfOSA, but is slower than Coreset.
>
>
> Q9: "How is $N_{query}^{ID}$ selected..."
>
> A9: Actually, the value of $un(x_{j}^u)$ represents the uncertainty, which comprehensively considers the instance's representativeness and informativeness. Therefore, we select the first b examples with the highest probability, as described in Line 155. We will give more details in the revision.

---

> > ### Comment · Reviewer_k1sx · 2023-08-11
> >
> > Thanks for your rebuttal! After carefully reading the author's response, my concerns have been well addressed. Thus, I have decided to raise my overall score, and I tend to recommend acceptance of this paper.

---

> > > ### Author Response · Authors · 2023-08-16
> > >
> > > Thank you for your helpful comments and feedback. Please let us know if there are further confusions/questions. We are happy to clarify and try to address them.

---

### Official Review · Reviewer_oop5 · 2023-07-05

**Soundness:** 3 good
**Presentation:** 3 good
**Contribution:** 3 good
**Rating:** 7
**Confidence:** 4

**Summary:**

This paper considers the open-set active learning problem, a sub-topic of active learning that focuses on non-iid settings. The authors constructed both an ID classifier and an OOD detector to implement open-set active learning. Specifically, they proposed a sampling scheme to ensure a balance of ID and OOD samples, which helps both the ID classifier and the OOD detector. The authors conducted a series of experiments, and the results of these experiments verified the effectiveness of the proposed method.

**Strengths:**

1. The considered problem is important and highly relevant to the machine learning community.
2. The proposal is simple and reasonable. The authors suggest that out-of-distribution (OOD) samples can also help the process of open-set active learning from the perspective of OOD detection. They use a simple and effective balanced sampling approach to alleviate the learning process differences and improve the performance of open-set active learning. This proposal suggests that actively annotating some OOD samples can enhance the OOD detector and help open-set active learning. This supports the claim of the title "Not All Out-of-Distribution Data Are Harmful" and extends the previous focus on mainly querying ID samples.
3. The experiments are sufficient, and the effectiveness of the proposal has been clearly verified.

**Weaknesses:**

CCAL is also a well-known open-set active learning method, which should be discussed and compared in the experiments.
[CCAL] Contrastive active learning under class distribution mismatch. TPAMI’22

It is regretful that there is no relevant theoretical analysis to further support this work.

Miscellaneous:
The results in Table 1 are too dense. Consider adjusting the layout.
Also, consider using the same text font in Figures 3, 4, 5, 6, and 7 as in the main body.

**Questions:**

The proposal uses meta-learning to calculate meta-weights for obtaining representative metrics of the samples. Compared to previous methods such as clustering and coreset, does this complex method introduce too much computational overhead? Does it have any special advantages? In the experiment, can we obtain good results using traditional representative metric methods?

---
Thanks for the clarifications. Most of the concerns have been addressed. I will keep my score.

**Limitations:**

The authors have provided a discussion on the broader impact.

---

> ### Author Rebuttal · Authors · 2023-08-09
>
> Q1: "CCAL is also a well-known open-set active learning method, which should..."
>
> A1: We have compared the proposed PAL with CCAL on CIFAR-10 and CIFAR-100 with the ID proportion of 20%. The results in Table 1 reveal that PAL outperforms CCAL, for the reason that CCAL does not actively use OOD data like the existing open-set AL method, which may lead to bias in instances' similarity calculation for unlabeled data.
>
> Table 1: Comparison of classification accuracy (%) for CCAL and PAL on CIFAR-10 and CIFAR-100 with an ID proportion of 20%.
>
> |       | CIFAR-10 |       |       |       |       | CIFAR-100 |       |       |       |       |
> | :---: | :------: | :---: | :---: | :---: | :---: | :-------: | :---: | :---: | :---: | :---: |
> | Round |    1     |   3   |   5   |   7   |   9   |     1     |   3   |   5   |   7   |   9   |
> | CCAL  |  88.90   | 94.35 | 96.20 | 97.55 | 97.70 |   45.10   | 50.90 | 53.40 | 57.20 | 60.45 |
> |   PAL    | **91.14** | **95.59** | **97.59** | **98.49** | **98.74** | **45.65** | **53.04** | **60.04** | **65.64** | **69.39** |
>
> Q2: "...no relevant theoretical analysis to further support..."
>
> A2: As mentioned in the **Global Rebuttal**, we have provided a theoretical analysis to support the proposed PAL method. The generalization results reveal that PAL has a better generalization error bound than the traditional AL method, showing the effectiveness of the proposed PAL. We will provide more comprehensive details about the theoretical analysis in the revision.
>
>
>
> Q3: "...results in Table 1 are too dense..."
>
> A3: Thank you. We will modify the table and improve the readability in the revision.
>
>
>
> Q4: "...does this complex method introduce too much computational overhead? Does it have any special advantages..."
>
> A4: Actually, the core idea of active learning is to control the budget of manual annotations to maximize the advancements in model performance. Although the introduction of meta-weights increases training costs, the performance has been improved by selecting more effective instances. Moreover, we have replaced the meta-weight with Coreset to validate the effectiveness, and the PAL with Coreset method, i.e., Core_PAL. The results in Table 2 show that Core_PAL performs worse than PAL, revealing the effectiveness of meta-weight in selecting instances.
>
>
> Table 2: Comparison of classification accuracy (%) for Core_PAL and PAL on CIFAR-10 and CIFAR-100 with an ID proportion of 20%.
>
> |          | CIFAR-10 |       |       |       |       | CIFAR-100 |       |       |       |       |
> | :------: | :------: | :---: | :---: | :---: | :---: | :-------: | :---: | :---: | :---: | :---: |
> |  Round   |    1     |   3   |   5   |   7   |   9   |     1     |   3   |   5   |   7   |   9   |
> | Core_PAL |  87.09   | 93.29 | 96.74 | 97.44 | 98.29 |   44.79   | 49.75 | 59.59 | 63.74 | 65.99 |
> |   PAL    | **91.14** | **95.59** | **97.59** | **98.49** | **98.74** | **45.65** | **53.04** | **60.04** | **65.64** | **69.39** |

---

> > ### Comment · Reviewer_oop5 · 2023-08-14
> >
> > Thank you for the clarifications. I agree that the purpose of active learning is to improve performance. However, in practical applications, facing large-scale unlabeled data pools, high computational costs can severely limit the practicality of algorithms. Therefore, could you provide a computational cost analysis of the PAL method, especially the meta-weight learning process, as well as theoretical convergence and empirical convergence rounds in experiments? These will help readers understand the computational costs and practicality behind the algorithm.

---

> > > ### Author Response · Authors · 2023-08-16
> > >
> > > Thank you for your valuable comments. To assess the computational costs, we calculated the components related to uncertainty, meta-weight, and the total cost on CIFAR-100 with an ID proportion of 20% for each round of total 10 rounds. Our findings indicate that the cost remains relatively stable throughout each round, and our method exhibits the ability to converge. Take the first round as an example, the total running time is 108.96 min, in which the running time of meta-weight is 71.21 min (including the running time of 70.92 min for $S^{ID}$) and the running time of classification is 37.75 min. Additionally, our approach can converge within 200 epochs in most rounds. As evident from the data presented in Table 5 of the accompanying single-page PDF, the proposed PAL is faster than most approaches, except for Coreset. This is due to the utilization of the more computationally intensive meta-weight approach for measuring sample representativeness, which results in improved performance. With some assumptions such as smoothness and bounded variance for the loss function of interest, one could show that the proposed method can achieve the convergence rate of $O(1/\sqrt{T})$, where $T$ is the number of iterations. We then empirically find that the Pearson correlation coefficient between training loss and $1/\sqrt{T}$ is 0.9581,  showing the training loss is highly linear correlated with $1/\sqrt{T}$, which matches the theoretical convergence rate. Thank you again for your feedback! We will add more experiments in the revised version. Please let us know if you have any additional questions. We are happy to provide additional experimental results.

---

> > > > ### Comment · Reviewer_oop5 · 2023-08-16
> > > >
> > > > Thank you for the clarifications and additional supporting results.

---

### Official Review · Reviewer_d3fV · 2023-07-05

**Soundness:** 3 good
**Presentation:** 3 good
**Contribution:** 3 good
**Rating:** 6
**Confidence:** 4

**Summary:**

In this paper, the authors aims to improve open-set active learning, where unlabelled set may contains some open-set instances. To do this, they propose a new sampling scheme, called progressive active learning. Specifically, they use a progressive sampling method to select valuable OOD data for the tradeoff between pseudo-ID and pseudo-OOD instances in each round. Experiments on various datasets are conducted to show the effectiveness of the proposed method.

**Strengths:**

1. This paper is well-writing and easy-to-understand. I believe readers can easily get the core idea of the proposed method.

2. The empirical performance is non-trivial. From Table 1, we can observe the improvement of the proposed method is significant.

3. The motivation is clear. I agree that OOD data should not be simply filtered out for active learning.

**Weaknesses:**

1. The method is somewhat heuristic and not novel enough. In my view, both OVA classifier and meta-weight are not original contribution of this paper. The technical novelty of the progressive sampling is not sufficient to support the acceptance in this conference.

2. The literature of leveraging OOD data to improve generalisation is missing, like OAT [1], ODNL [2] and Open-sampling [3]. The authors may need to discuss the relationship between the proposed method and these works.



[1] Lee, Saehyung, et al. "Removing Undesirable Feature Contributions Using Out-of-Distribution Data.", ICLR, 2021.

[2] Wei, et al. Open-set Label Noise Can Improve Robustness Against Inherent Label Noise. NeurIPS 2021

[3] Wei, et al. Open-Sampling: Exploring Out-of-Distribution data for Re-balancing Long-tailed datasets. ICML 2022.

---------

After reading the responses from the authors, my concerns have been well addressed, so I lean towards acceptance for this paper.

**Questions:**

Do you mean the positive effect of OOD data is only in detecting OOD examples? You write "improving the ID purity in query sets" in line 66.

---

> ### Author Rebuttal · Authors · 2023-08-09
>
> Q1: " ...The technical novelty of the progressive sampling is not sufficient..."
>
> A1: Open-set active learning aims to strategically select pure ID data and filter out OOD data, necessitating the presence of a robust OOD detector. The common practice of traditional open-set active learning methods, as depicted in Figure 1 (b) in the manuscript, usually relies on detected ID data to build the OOD detector. Subsequently, these methods primarily focus on selecting pseudo-ID instances, which would lead to the weakening of the OOD detector due to the lack of active selection of effective OOD data and affects the ID purity during the selection process. To address this challenge, we propose a simple yet effective sampling scheme that progressively selects pseudo-ID and pseudo-OOD instances in each round, especially adding valuable OOD data in the initial round. Thereby, it could enhance the capacity of the OOD detector and simultaneously promote the ID classifier by increasing the ID purity. We will highlight the contributions in the revision.
>
> On the theoretical side, we provide a theoretical analysis of generalization error, which is commonly used in the literature of learning theory. Our theoretical analysis reveals that the proposed PAL has a better generalization error bound than the standard AL without using detected OOD data, showing the effectiveness of the proposed PAL. Please refer to **Global Response** for more details about the theoretical analysis.
>
>
> Q2: "... literature of leveraging OOD data to improve generalization is missing..."
>
> A2: Thanks for your suggestion. Unlike OAT, ODNL, and Open-sampling, which directly utilize OOD data to improve the ID classifier, our approach focuses on enhancing the OOD detector using detected OOD data, thereby controlling the ID purity and improving the ID classifier. We will include more discussion in the revision.
>
>
> Q3: "Do you mean the positive effect of OOD data is only in detecting OOD examples..."
>
> A3: Sorry for the confusion. The positive effect of selected OOD data can not only promote the OOD detector building for better detecting OOD data in subsequent phases but also improve the ID purity. We will make it clear and give more details in the revision.

---

> > ### Comment · Reviewer_d3fV · 2023-08-11
> >
> > Thank you for the detailed response. My concerns have been well addressed, so I lean towards acceptance for this paper.

---

> > > ### Author Response · Authors · 2023-08-16
> > >
> > > Thank you for your helpful comments and feedback. Please let us know if there are further confusions/questions. We are happy to clarify and try to address them.

---

### Author Rebuttal · Authors · 2023-08-09

We sincerely thank the PC, SAC, ACs, and reviewers for handling and reviewing our paper. All constructive and valuable comments are helpful in further improving our paper.

Since most reviewers have mentioned the lack of theoretical analysis, we provided a generalization analysis. Our theoretical results reveal that the proposed PAL has a better generalization error bound than traditional AL, showing the effectiveness of the proposed PAL. We include the details as follows.

We also include additional experimental results in attached ***one-page PDF*** file.


======== Theoretical Analysis ========

To theoretically understand the use of detected OOD data in the OOD detector, we first give the following notations. Suppose that the samples $(\mathbf{x},y)\in(\mathcal{X},\mathcal{Y})$ follows a unknown distribution $\mathcal{P}$, where $\mathcal{X} = \mathbf{R}^d$ is input space and $\mathcal{Y} = \\{0,1\\}$ is output space. The label of ID data is $y=0$, and the label of OOD data is $y=1$. Let $\ell:\mathbf{R}\times\mathcal{Y}\longrightarrow \mathbf{R}^+$ be the loss of interest. With the input $\mathbf{x}$ and its corresponding label $y$, the expected loss is $\mathcal{L}(f) = \mathbb{E}_{\mathcal{P}}[\ell(f(\mathbf{x}),y)]$.

Suppose we have a training dataset $\\{(\mathbf{x_1},y_1), \dots, (\mathbf{x_n},y_n)\\}$ drawn from distribution $\mathcal{P}$, then the empirical loss is $\mathcal{\widehat L}(f) = \frac{1}{n}\sum_{i=1}^{n}\ell(f(\mathbf{x_i}),y_i)$. Let $C(\mathcal{F})$ be some proper complexity measure of the family of hypothesis class $\mathcal{F}$. We assume that the loss is Lipschitz with constant $L$ and the Rademacher complexity $\mathfrak{R}_n(\mathcal{F}) \le \sqrt{\frac{C(\mathcal{F})}{n}}$. To simplify the analysis, we assume the distribution of ID and OOD samples is balanced and the number of OOD samples is the same as the number of ID samples in the training set. We suppose that the number of detected ID data is $m$ while the number of detected OOD data is $n-m$ where $m<n$, and among the detected ID data, the number of real ID data is $m_0 := \alpha m$ while the number of real OOD data is $m_1 := (1-\alpha)m$  with $0<\alpha<1$.

**Theorem 1.** For a Lipschitz loss $\ell$ bounded by $c$, we have the following results with probability at least $1-\delta$ simultaneously.

For the proposed PAL method, the generalization error bound is
\begin{align}
    \mathcal{L}(f_{PAL})- \mathcal{\widehat L}(f_{PAL}) \le 2L\sqrt{\frac{C(\mathcal{F})}{n}} + c\sqrt{\frac{\log(1/\delta)}{2n}} \le O\left(\frac{1}{\sqrt{n}}\right).
\end{align}
For the standard AL method, the generalization error bound is

\begin{align}
    \nonumber \mathcal{L}(f_{AL})- \mathcal{\widehat L}(f_{AL}) \le & L\sqrt{\frac{C(\mathcal{F})}{\alpha m}} + L\sqrt{\frac{C(\mathcal{F})}{(1-\alpha)m}} + \frac{c}{2}\sqrt{\frac{\log(2/\delta)}{2\alpha m}} + \frac{c}{2}\sqrt{\frac{\log(2/\delta)}{2(1-\alpha)m}} \le O\left(\frac{1}{\sqrt{\alpha m}} + \frac{1}{\sqrt{(1-\alpha)m}}\right),
\end{align}
where $0<\alpha<1$ and $n<m$.

**Remark.**   Since $0<\alpha<1$ and $n<m$,  we know $\frac{1}{\sqrt{n}} \le \frac{1}{\sqrt{\alpha m}} + \frac{1}{\sqrt{(1-\alpha)m}} $, showing that the generalization error bound for PAL method is better than the bound for standard AL method. That is to say, the use of detected OOD data can improve the effectiveness of the OOD detector.

**Proof Sketch.** First, we proof the generalization error of the proposed PAL method. Following Theorem 1 of [R1], we have
\begin{align}
    \nonumber \mathcal{L}(f_{PAL})- \mathcal{\widehat L}(f_{PAL}) \le 2L\mathfrak{R}_n(\mathcal{F})+ c\sqrt{\frac{\log(1/\delta)}{2n}},
\end{align}
where $\mathfrak{R}_n(\mathcal{F})$ is the Rademacher complexity of a function class $\mathcal{F}$ and $n$ is the number of training samples. Withe the Rademacher complexity $\mathfrak{R_n}(\mathcal{F}) \le \sqrt{\frac{C(\mathcal{F})}{n}}$, we give equation (1).

Next, we will proof the generalization error of the standard AL method. To this end, we denote by $\mathcal{P} = \mathcal{P}(\mathbf{x}|y=0)$ the conditional probability for ID data and $\mathcal{P_1} = \mathcal{P}(\mathbf{x}|y=1)$ the conditional probability for OOD data. Let $\mathcal{L_j}(f)$ be the loss from class $j \in \\{0,1\\}$: $\mathcal{L_j}(f) = \mathbb{E_{\mathcal{P_j}}}[\ell(f(\mathbf{x}),y)]$, and let $\mathcal{\widehat L_j}(f)$ be its corresponding empirical loss.
Then by applying the standard analysis for each class $j$ in Theorem 1 of [R1], with probability $1-\delta/2$ we have
\begin{align}
    \nonumber \mathcal{L_j}(f_{AL})- \mathcal{\widehat L_j}(f_{AL}) \le 2L\mathfrak{R_{m_j}}(\mathcal{F})+ c\sqrt{\frac{\log(2/\delta)}{2m_j}}, (j = 0,1).
\end{align}

Since the distribution of ID and OOD samples is balanced, we have
$\mathcal{L}(f_{AL}) = \frac{1}{2} \mathcal{L_0}(f_{AL}) + \frac{1}{2} \mathcal{L_1}(f_{AL})$ due to the definitions of the loss functions. Similarly, due to the number of OOD samples is same as the number of ID samples in training set, we know $\mathcal{\widehat L}(f_{AL}) = \frac{1}{2}\mathcal{\widehat L_0}(f_{AL}) + \frac{1}{2}\mathcal{\widehat L_1}(f_{AL})$. Finally, by applying the union bound, we have
\begin{align}
    \nonumber \mathcal{L}(f_{AL})- \mathcal{\widehat L}(f_{AL}) \le L\mathfrak{R_{m_0}}(\mathcal{F})+ + L\mathfrak{R_{m_1}}(\mathcal{F})
    +\frac{c}{2}\sqrt{\frac{\log(2/\delta)}{2m_0}} + \frac{c}{2}\sqrt{\frac{\log(2/\delta)}{2m_1}}.
\end{align}

With the Rademacher complexity $\mathfrak{R}_{m_j}(\mathcal{F}) \le \sqrt{\frac{C(\mathcal{F})}{m_j}}$ for $j=0,1$, we complete the proof of equation (2) by plugging in $m_0 = \alpha m$ and $m_1=(1-\alpha)m$.

[R1] Sham M Kakade, Karthik Sridharan, and Ambuj Tewari. On the Complexity of Linear Prediction: Risk Bounds, Margin Bounds, and Regularization. In Advances in Neural Information Processing Systems, pages 793–800, 2009.

---

### Decision · Program_Chairs · 2023-09-21

**Decision:**

Accept (poster)

**Comment:**

This paper was reviewed by four experts and received one Borderline Accept, two Weak Accept and one Accept as the ratings. All the reviewers agreed that the paper addressed a problem relevant to the machine learning community; it is well-written, and the ideas are easy to understand. They have also mentioned that the empirical results are encouraging and show that the performance improvement achieved by the proposed method is significant.

The reviewers had raised concerns about the theoretical analysis of the method, which was provided by the authors in the rebuttal. Concerns were also raised about the novelty of the paper and its computation cost, which were answered convincingly by the authors in the rebuttal. One of the reviewers had mentioned that the proposed technique should be compared against CCAL, a well-known method for open-set active learning; the authors have conducted additional experiments, and reported the results in the rebuttal.

The reviewers, in general, have a positive opinion about the paper and its contributions. Most of them have been satisfied with the author rebuttal, have raised their scores and have recommended acceptance. Based on the reviewers’ feedback, the decision is to recommend the paper for acceptance to NeurIPS 2023. The reviewers have provided some valuable suggestions, particularly about the discussion on open-set SSL in the Related Work section and a clear description of the loss function used for training. The authors are encouraged to address these in the final version of their paper. We congratulate the authors on the acceptance of their paper!